# Tumor-selective catalytic nanomedicine by nanocatalyst delivery

Minfeng Huo[1,2], Liying Wang[1,2,3], Yu Chen [1] & Jianlin Shi [1]

Tumor cells metabolize in distinct pathways compared with most normal tissue cells. The resulting tumor microenvironment would provide characteristic physiochemical conditions for selective tumor modalities. Here we introduce a concept of sequential catalytic nanomedicine for efficient tumor therapy by designing and delivering biocompatible nanocatalysts into tumor sites. Natural glucose oxidase (GOD, enzyme catalyst) and ultrasmall $Fe_3O_4$ nanoparticles (inorganic nanozyme, Fenton reaction catalyst) have been integrated into the large pore-sized and biodegradable dendritic silica nanoparticles to fabricate the sequential nanocatalyst. GOD in sequential nanocatalyst could effectively deplete glucose in tumor cells, and meanwhile produce a considerable amount of $H_2O_2$ for subsequent Fenton-like reaction catalyzed by $Fe_3O_4$ nanoparticles in response to mild acidic tumor microenvironment. Highly toxic hydroxyl radicals are generated through these sequential catalytic reactions to trigger the apoptosis and death of tumor cells. The current work manifests a proof of concept of catalytic nanomedicine by approaching selectivity and efficiency concurrently for tumor therapeutics.

---

[1] The State Key Laboratory of High Performance Ceramic and Superfine Microstructures, Shanghai Institute of Ceramics, Chinese Academy of Sciences, Shanghai 200050, China. [2] University of Chinese Academy of Sciences, Beijing 100049, China. [3] School of Physical Science and Technology, ShanghaiTech University, Shanghai 201210, China. Correspondence and requests for materials should be addressed to Y.C. (email: chenyu@mail.sic.ac.cn) or to J.S. (email: jlshi@mail.sic.ac.cn)

Chemotherapy, the most conventional modality for cancer therapy, suffers from severe side effects on account of the low bioavailability and tumor-therapeutic specificity. A number of physical therapeutic modalities, such as radiation[1, 2], ultrasound[3, 4], photoacoustic/photothermal[5, 6] or microwave therapies[7, 8] are capable of positioning the therapeutic sites by imaging-guidance, but these strategies may also cause severe damages to surrounding normal tissues and/or induce undesired tumor metastasis. To achieve a more focused and tumor-specific therapy, tumor microenvironment (TME) has been broadly exploited recently, in which the cellular metabolism, biosynthetic intermediates and physical environment are prominently different from those in normal tissues[9–11]. TME-responsive drug releasing and diagnostic imaging have been extensively developed[12–15], however, the introduction of toxic anticancer agents would still induce undesired distribution to normal organisms and damage these tissues subsequently.

Hence, it is here conceived that, if the intrinsic substances in the TME, which are naturally non-toxic and biocompatible, could be in situ transformed into toxic agents against tumors with significant therapeutic efficacy based on the specific features of TME, under the stimuli of intratumor-delivered non-toxic agent (s) or pro-catalyst(s), the tumor therapy will be realized without or with minimized harmful side effects. This strategy, if applicable, can only trigger the therapeutic process within tumorous tissues rather than other normal ones, hopefully resulting in the concurrent high therapeutic efficacy and negligible damages to normal tissues and/or organs.

Mesoporous materials are born as excellent catalyst supports, which enable highly dispersed catalytic active sites loading in the mesostructure to guarantee excellent catalytic performances and the free diffusion of reactants and/or products[16, 17]. Especially, mesoporous silica-based versatile nanosystems (e.g., mesoporous silica nanoparticles, designated as MSNs) have been extensively demonstrated to be biocompatible both in vitro and in vivo[18–20]. Therefore, MSNs are regarded as one of the most favorable catalyst supports for possible catalytic reaction-based biomedical applications.

The following consideration is associated with the choice of adequate catalytic reactions for efficient tumor therapy, based on which the corresponding nanocatalysts can be determined and selected. Ferromagnetic nanoparticles ($\gamma$-Fe$_2$O$_3$ or Fe$_3$O$_4$ NPs)

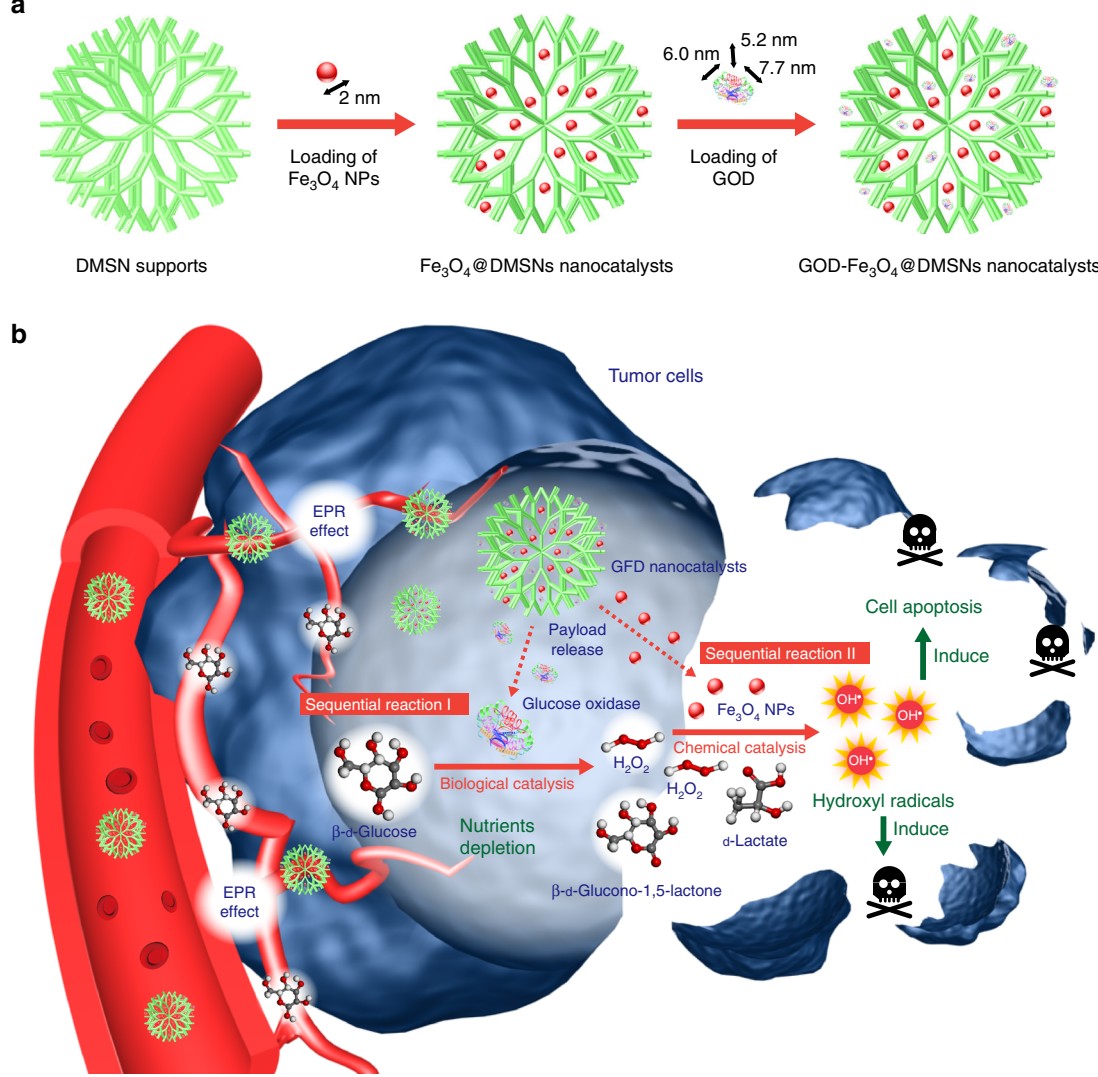

**Fig. 1** Fabrication and catalytic-therapeutic schematics of sequential GFD NCs. **a** Synthetic procedure for Fe$_3$O$_4$@DMSNs nanocatalysts and GOD-Fe$_3$O$_4$@DMSNs nanocatalysts. The sizes of the prepared Fe$_3$O$_4$ nanoparticles and adopted GOD are indicated in the scheme. **b** The scheme of sequential catalytic-therapeutic mechanism of GFD NCs on the generation of hydroxyl radicals for cancer therapy

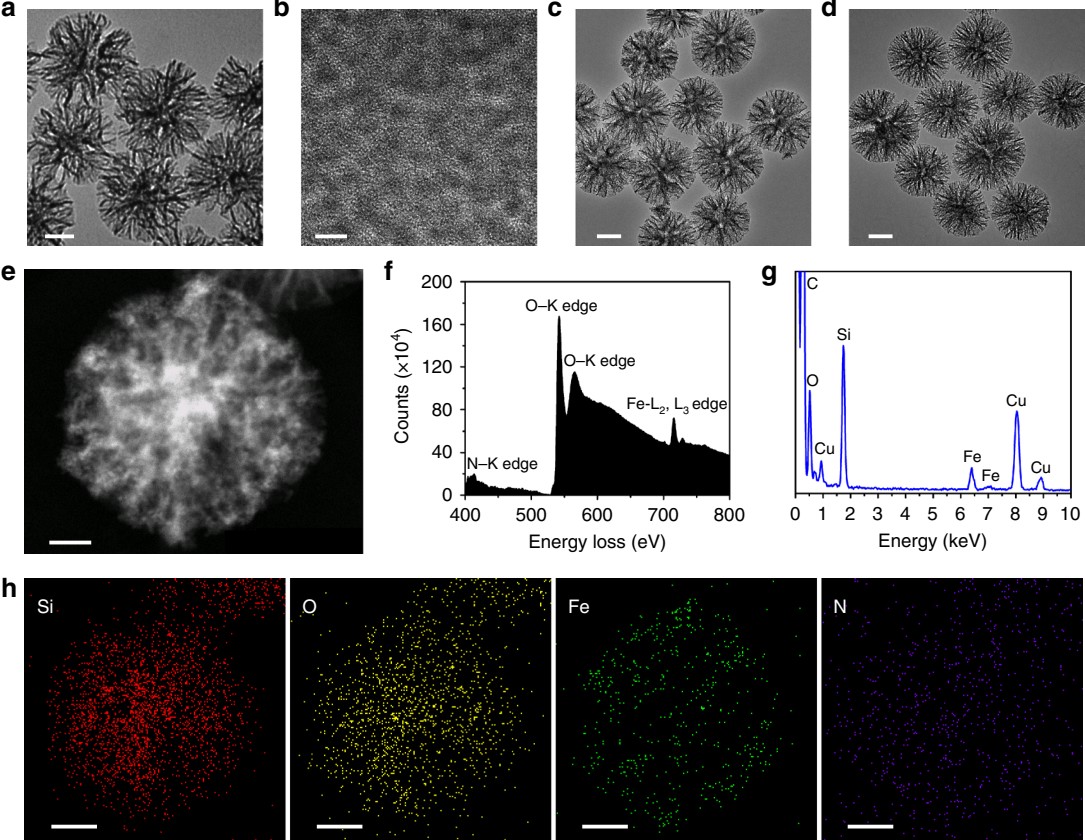

**Fig. 2** Structural and compositional characterizations of DMSN supports, FD NCs, and GFD NCs. TEM images of **a** DMSN supports, **b** ultrasmall $Fe_3O_4$ NPs, **c** FD NCs, and **d** GFD NCs. *Scale bar*: **a**, **c**, **d**: 100 nm; **b**: 5 nm. **e** Dark-field image, **f** EELS, **g** EDS, and **h** corresponding area-elemental mappings of the sequential GFD NCs. *Scale bar*: 50 nm

have been revealed to perform dual enzyme-like activity both in vitro and in vivo in a pH-dependent manner. These iron oxide nanoparticles (IONPs) could catalytically decompose $H_2O_2$ into non-toxic $H_2O$ and $O_2$ under neutral pH condition, presenting catalase-like activity. More interestingly, they could disproportionate $H_2O_2$ into highly toxic reactive oxygen species (ROS) - hydroxyl radicals (·OH), displaying peroxidase-like activity under acidic condition[21, 22]. Therefore, IONPs are considered as potential tumor-therapeutic nanozymes because such a site-specific generation of the hydroxyl radicals could induce the apoptosis and death of cancer cells under the mildly acidic microenvironment of tumor, leaving the normal cells unharmed[23]. However, the intracellular $H_2O_2$ level in tumor cells is too low for IONPs to generate high enough amount of hydroxyl radicals to produce satisfactory catalytic performance even under the catalysis of $Fe_3O_4$ NPs. Therefore, a strategy to elevate the intratumoral $H_2O_2$ level is to be developed.

Here, we show a biodegradable and sequentially functioning nanocatalyst (sequential nanocatalyst in the following) for the efficient catalytic tumor suppression via TME-responsive sequential catalytic reactions with high tumor specificity and therapeutic efficacy. Serving as the starting enzyme catalyst, natural glucose oxidase (GOD) is competent to catalyze the glucose in tumor region into abundant $H_2O_2$. The elevated $H_2O_2$ is then catalyzed by the downstream $Fe_3O_4$ NPs via Fenton-like reactions to liberate highly toxic hydroxyl radicals, which could further induce tumor apoptosis and death. The as-designed biocompatible sequential nanocatalyst manifests a proof of concept of catalytic nanomedicine in the selective and effective tumor therapy by directly introducing non-toxic nano-sized catalysts into tumor.

## Results

**Catalytic-therapeutic mechanism of sequential GFD NCs.** The design and fabrication of the sequential nanocatalyst are illustrated in Fig. 1a. GOD, (size: 6.0 nm × 5.2 nm × 7.7 nm)[24] and synthetic ultrasmall $Fe_3O_4$ NPs (size: 2 nm)[25, 26] were successively integrated into the large mesopores (~40 nm) of dendritic MSN (DMSNs)[27] to form a composite nanocatalyst for sequential catalytic reactions directly within the tumor tissue, designated as GOD-$Fe_3O_4$@DMSNs nanocatalysts (GFD NCs). Such a sequential nanocatalyst features high biocompatibility because the DMSNs supports, GOD and $Fe_3O_4$ NPs are highly chemically compatible with each other, guaranteeing their biosafety in in vivo applications for catalytic tumor therapy.

The essential catalytic therapeutic concept of GFD NCs is based on the fast tumor metabolism and the resulting specific TME. Most tumor cells could transport and metabolize the nutrients at considerably faster rates than normal tissues through extensively constructed intratumoral vascular system to support the proliferation[28–30]. Among these nutrients, β-D-glucose is evidenced to be the most essential one judging from the characteristics of tumor metabolism. These tumor cells could eagerly progress low ATP-productive anaerobic glycolysis (2 moles ATP produced by 1 mole glucose), without further oxidative phosphorylation, a higher ATP-productive process in normal proliferative cells (32 moles ATP per mole glucose)[31]. This glycolytic character of tumor cells leads to the exaggerated demand for glucose nutrients for tumor growth. Furthermore, overexpression of glucose transporters, including facilitative glucose transporters (GLUTs) and sodium-dependent glucose transporters (SGLTs), has been found in tumor cells[32, 33]. Based

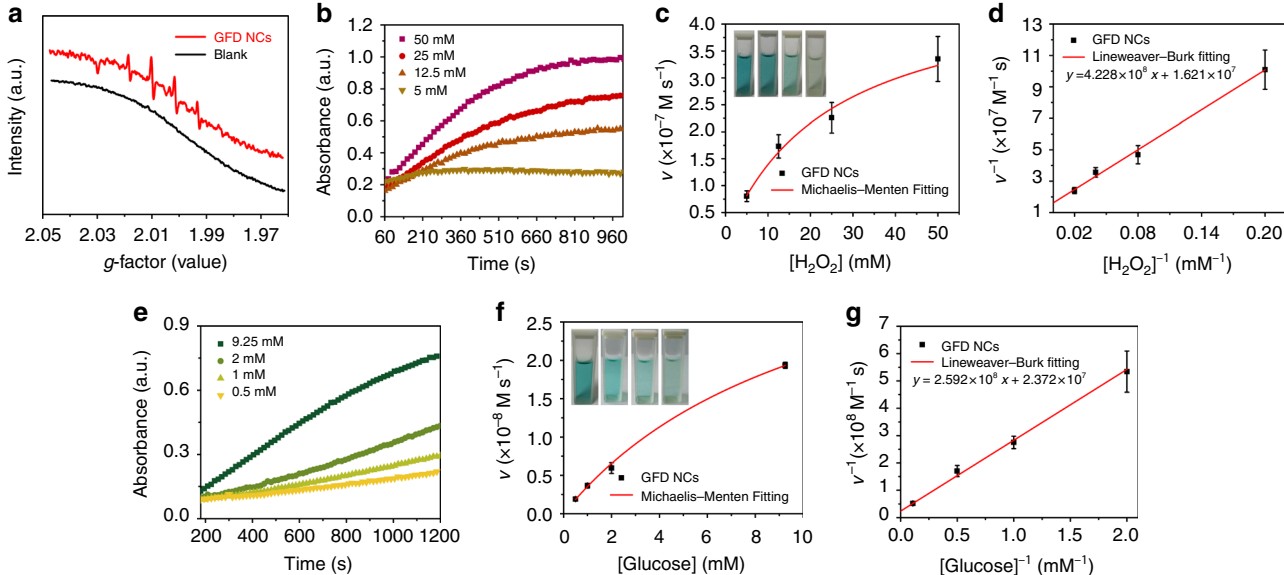

**Fig. 3** In vitro characterizations of the catalytic performances of sequential GFD NCs. **a** ESR spectra of GFD NCs (10 μg ml⁻¹) with or without the addition 10 mM glucose. Michaelis–Menten steady-state kinetics of (**b–d**) $Fe_3O_4$ NPs and (**e–g**) sequential GOD-$Fe_3O_4$ NPs confined in GFD NCs. Time-course absorbance of GFD NCs upon the addition of varied concentrations of (**b**) $H_2O_2$ (50, 25, 12.5, and 5 mM) or (**e**) β-D-glucose (9.25, 2, 1, and 0.5 mM). Michaelis–Menten kinetics and Lineweaver–Burk plotting of (**c**), (**d**) $Fe_3O_4$ NPs and (**f**), (**g**) sequential GOD-$Fe_3O_4$ NPs confined within the mesopores of DMSN. Mean values and error bars are defined as mean and s.d., respectively. The initial velocities were calculated in eight periods (15 s per period) in one experiment

on the promoted glycolysis (Warburg effect) and glucose-transporting facilities, tumor cells are exceptionally reliant on the glucose nutrient. On the other aspect, under the elevated level of active lactate dehydrogenase, most pyruvate molecules will be catalyzed into lactate molecules, inducing mild acidic environment (pH of around 6.0) in the tumorous cytoplasm, known as lactate acidosis[34, 35].

The as-designed GFD NCs could therefore function in sequential biological–chemical catalytic reactions toward efficient catalytic tumor therapy based on the specific features of TME (Fig. 1b). Initially, the GFD NCs could penetrate into tumor tissue via the typical enhanced permeability and retention (EPR) effect. The GOD in GFD NCs could effectively deplete the glucose nutrients by an enzyme-catalyzed bioreaction, which also in situ generates abundant $H_2O_2$ molecules (Fig. 1b, Sequential reaction I). Hopefully, these post-generated $H_2O_2$ molecules, acting as the substrate, could be further catalyzed to produce large amounts of highly toxic hydroxyl radicals by the co-encapsulated ultrasmall $Fe_3O_4$ nanozymes in GFD NCs via a Fenton-like reaction, which could effectively induce the cell apoptosis and death (Fig. 1b, Sequential reaction II). Therefore, the designed sequential GFD NCs could double-function specifically based on their unique catalytic performance, i.e., depleting glucose to starve the cancer cells, and elevating the $H_2O_2$ levels in tumor tissue for promoting the peroxidase-like Fenton reaction to produce large amounts of hydroxyl radicals for tumor therapy (Fig. 1b).

**Structure and compositions of GFD NCs.** Transmission electron microscopic (TEM) images of the as-synthesized DMSN supports show a uniform spherical morphology with dendrimer-like nanostructure (Fig. 2a). The dendritic porous structure provides large mesopores as the reservoirs for the encapsulation of guest catalysts such as $Fe_3O_4$ nanozymes and/or GOD enzymes. Both hydrophobic ultrasmall $Fe_3O_4$ NPs of ~2 nm in diameter (Fig. 2b, Supplementary Fig. 1) and GOD could be effectively collected and accumulated into the mesopores successively by DMSNs to form FD NCs (Fig. 2c) and subsequent GFD NCs (Fig. 2d). Scanning electron microscopic images of DMSN supports, FD NCs and GFD NCs (Supplementary

Fig. 2) present that the dendritic nanostructure of DMSN has been perfectly kept after $Fe_3O_4$ NPs and GOD encapsulations.

To execute precise elemental analysis of encapsulated catalysts, characterizations by dark-field images (DFI), electron energy loss spectroscopy (EELS), energy dispersive spectrometer (EDS), and corresponding elemental mapping (EDS-mapping) of GFD NCs (Fig. 2e–k) and FD NCs (Supplementary Fig. 3) were performed. The DFIs clearly show the dendritic pore structure of both GFD NCs (Fig. 2e) and FD NCs (Supplementary Fig. 3a), in which high-Z $Fe_3O_4$ NPs are supposed to distribute centrally in the relatively bright area of both GFD NCs and FD NCs. A distinctive N–K edge is detected in the EELS spectrum of GFD NCs (Fig. 2f) rather than in that of FD NCs (Supplementary Fig. 3b), elucidating the successful encapsulation of GOD. Furthermore, the coexistence of Si, O, Fe signals is found in the EDS spectra of both GFD NCs (Fig. 2g) and FD NCs (Supplementary Fig. 3c), and the homogeneous distributions of Si, O, Fe, N in GFD NCs are confirmed by the EDS-elemental mapping (Fig. 2h), indicating the successful and dispersive loadings of $Fe_3O_4$ NPs and GOD simultaneously. The EDS-mapping of FD NCs also reveal the homogeneous distributions of Si, O, Fe elements, illustrating the successful and dispersive loading of $Fe_3O_4$ NPs in FD NCs as well (Supplementary Fig. 3d–f).

The specific surface area (SSA) and the pore diameter of DMSN supports are determined by $N_2$ absorption–desorption technique using Brunauer–Emmett–Teller and Barrett–Joyner–Halenda methods, respectively. DMSN show a SSA of as high as 673.8 m² g⁻¹ and an average pore diameter of 40 nm (Supplementary Fig. 4a, b). As an excellent nanocollector, DMSN is capable of encapsulating large enzymes, conquering the critical issue of rather low capacities of traditional MSNs (~5 nm) in loading molecules and/or nanoparticles of larger dimensions. The large pore-sized DMSN could effectively collect the ultrasmall $Fe_3O_4$ nanoparticles into the mesopores at the initial enzyme-loading process. The hydrophobic $Fe_3O_4$ nanoparticles are firmly confined within the mesopores in the following GOD-loading process in an aqueous medium. The dynamic light scattering (DLS) data acquired in aqueous solutions demonstrate that the

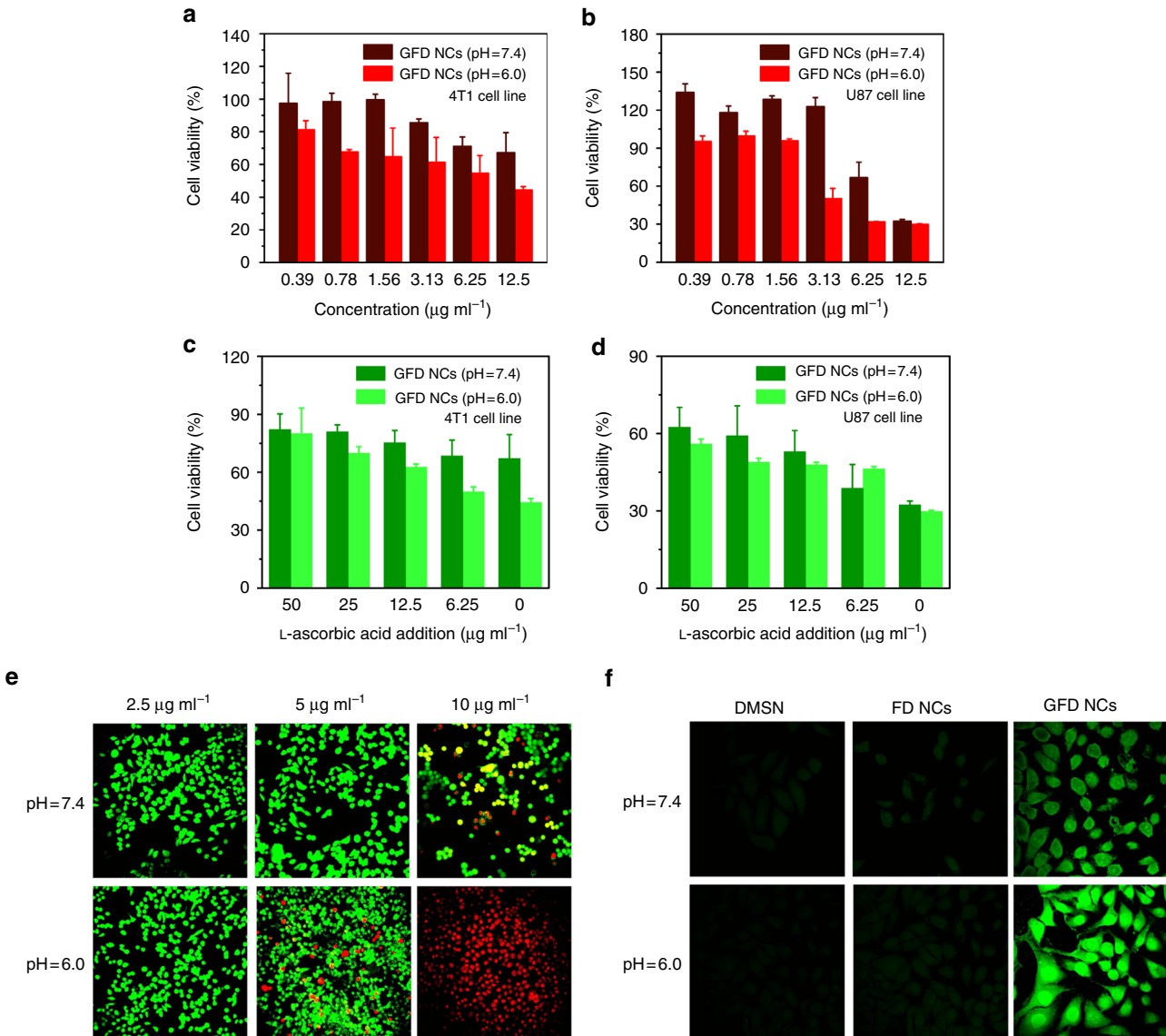

**Fig. 4** In vitro cytotoxicity profiles and intracellular catalytic mechanism. In vitro 4T1 (**a**) and U87 (**b**) cytotoxicity profiles of GFD NCs under neutral (pH = 7.4) and acidic (pH = 6.0) conditions. L-ascorbic acid-assisted cell rescue profiles of (**c**) 4T1 and (**d**) U87 cancer cells' cytotoxicity induced by 12.5 μg ml$^{-1}$ of GFD NCs. Mean values and error bars are defined as mean and s.d., respectively. Experiments were performed in quadruplicate. **e** CLSM images of viable and dead cell distributions after co-incubation with GFD NCs at varied concentrations: 2.5, 5, and 10 μg ml$^{-1}$ under neutral (pH = 7.4) and acidic (pH = 6.0) conditions for 4 h and subsequently stained with calcein-AM/PI solutions. **f** CLSM images of 4T1 cells after co-incubation with DMSN, FD NCs, and GFD NCs under neutral (pH = 7.4) and acidic (pH = 6.0) conditions for 4 h and subsequently stained with ROS fluorescence probe DCFH-DA

average hydrodynamic diameters ($D_h$) of DMSN, FD NCs, and GFD NCs are respectively at around 237.6, 319.6, and 255.3 nm (Supplementary Fig. 4c), and the zeta potentials are elevated from −45.6 to −32.0 mV and −27.6 mV, correspondingly (Supplementary Fig. 4d). Thermogravimetric (TG) and inductively coupled plasma-optical emission spectrometry (ICP-OES) measurements were used to determine the loading amounts of the GOD and Fe$_3$O$_4$ NPs in GFD NCs, which are 16.61% as calculated from TG curve of FD NCs and GFD NCs, and 15.87% as determined by ICP-OES of GFD NCs, respectively.

**In vitro catalytic performance of sequential GFD NCs**. As illustrated in the above section on the mechanism of catalytic therapy (Fig. 1b), GFD NCs initially catalyze β-D-glucose into β-D-glucono-1,5-lactone and H$_2$O$_2$ by the encapsulated GOD biologically. Sequentially, Fe$_3$O$_4$ NPs catalyze the disproportionation of H$_2$O$_2$ intermediate to generate highly toxic hydroxyl radicals (·OH) under tumorous acidic pH environment, while produce non-toxic

H$_2$O and O$_2$ under neutral pH environment. In order to identify the generation of radical species, 100 mM of typical nitrogen trap, 5,5-dimethyl-1-pyrroline-*N*-oxide (DMPO) was applied to trap short-lived radicals to form relatively long-lived radical-DMPO adducts. These adducts were then detected and identified by electron spin resonance (ESR) spectroscopy. It is important to note that the addition of glucose (50 mM) into GFD NCs solution (10 μg ml$^{-1}$) under mildly acidic condition (pH = 5.2) has generated considerable amount of hydroxyl radicals, as demonstrated by the presence of characteristic 1 : 2 : 2 : 1 hydroxyl radical signals[36] in ESR spectrum (Fig. 3a). Comparatively, no signals are observed in ESR spectrum under identical measurement condition without glucose addition. This result demonstrates the glucose-initiated and acidity-responsive generation of hydroxyl radicals under the catalysis by the introduced sequential GFD NCs.

To further clarify the functionalities of the delivered catalytic moieties, 3,3′,5,5′-tetramethyl-benzidine (TMB) was applied to

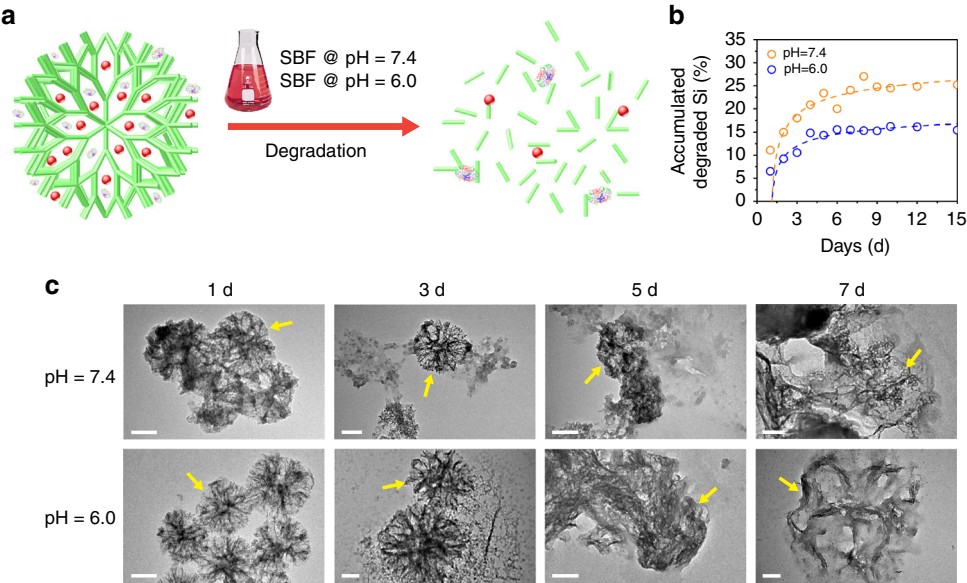

**Fig. 5** Biodegradation behavior of GFD NCs. **a** Schematic illustration of biodegradation process of GFD NCs. **b** Accumulated concentrations of released Si-based biodegradation products in SBF under neutral (pH = 7.4) and acidic (pH = 6.0) conditions. **c** TEM images of GFD NCs after biodegradation in neutral (pH = 7.4) and acidic (pH = 6.0) SBFs for varied durations: 1, 3, 5, and 7 days. *Scale bar*: 100 nm

monitor the radical production. The produced hydroxyl radicals from the disproportionation of $H_2O_2$ under the catalysis by $Fe_3O_4$ NPs in acidic environment will oxidize colorless TMB to chromogenic TMB cation-free radicals, which can be assayed at 650 nm ($E = 3.9 \times 10^4$ M$^{-1}$ cm$^{-1}$) in spectrometer. Based on the aforementioned principles, typical Michaelis–Menten steady-state kinetics were applied to investigate the catalytic functionalities and performances of the inorganic and biological enzymes confined in sequential GFD NCs.

$H_2O_2$ (50, 25, 12.5, and 5 mM) and glucose (9.25, 2, 1, and 0.5 mM) were served as the reactants in the assays of 100 µg ml$^{-1}$ GFD NCs, respectively. With $H_2O_2$ supply, the $Fe_3O_4$ NPs would solely function while maintaining GOD non-functioned. Hence, Michaelis–Menten steady-state kinetics of $Fe_3O_4$ NPs in GFD NCs was determined (Fig. 3b–d). The time-course absorbance upon the addition of $H_2O_2$ into GFD NCs in 20 mM sodium acetate buffer (pH = 5.2) are plotted in Fig. 3b and the corresponding average initial velocities are calculated. All average initial velocities of absorbance changes would then be converted as initial velocities ($v_0$) of cation-free radical production or hydroxyl radical formation via the Beer–Lambert law (Eq. 1), which were then plotted against the corresponding concentration and fitted with Michaelis–Menten curves (Eq. 2, Fig. 3c). Photographs in the *inset* of Fig. 3c visually present the chromogenic changes after 1000 s upon $H_2O_2$ additions of varied concentrations. Moreover, to determine the Michaelis–Menten constant ($K_M$) and maximum velocity ($V_{max}$), a linear double-reciprocal plot (Lineweaver–Burk plot, Eq. 3) was obtained as given in Fig. 3d. The $K_M$ and $V_{max}$ values were calculated to be 26.08 mM and $6.17 \times 10^{-8}$ M s$^{-1}$ for $Fe_3O_4$ NPs confined within the large mesopores of DMSN.

$$A = kbc \qquad (1)$$

$$v_0 = \frac{V_{max} \cdot [S]}{K_M + [S]} \qquad (2)$$

$$\frac{1}{v_0} = \frac{K_M}{V_{max}} \cdot \frac{1}{[S]} + \frac{1}{V_{max}} \qquad (3)$$

Similarly, in the subsequent assay for GFD NCs with glucose supply, the steady-state kinetics of sequential GFD NCs was investigated. The time-course absorbance was plotted in Fig. 3e and the initial velocities upon corresponding glucose addition were obtained and plotted following Michaelis–Menten and Lineweaver–Burk (Fig. 3f, g) equations. The Michaelis–Menten constant and the maximum velocities of sequential GOD-$Fe_3O_4$ NPs were determined to be 10.93 mM and $4.22 \times 10^{-8}$ M s$^{-1}$.

Compared to the naked enzymes, the sequential GFD NCs show slow but preferable steady-state kinetics judging from the $K_M$ and $V_{max}$ values[37]. The $K_M$ value for GOD in GFD NCs demonstrates that the sequential nanocatalysts would give 50% of maximum catalytic activity under 10.93 mM β-D-glucose. From this result, it can be known that the GFD NCs could perform sufficient therapeutic effect toward most of the cancer cells since the endogenous glucose concentrations in the cytoplasm of tumor cells are usually far below 20 mM[38]. Importantly, once all active sites of the sequential nanocatalyst were occupied, GFD NCs could catalyze glucose nutrients at the maximum velocity of $4.22 \times 10^{-8}$ M s$^{-1}$, providing moderate and steady therapeutic effects against cancer.

**In vitro cytotoxicity profiles**. The aforementioned in vitro investigation has demonstrated the efficient production of hydroxyl radicals by sequential GFD NCs. To gain an insight into the cytotoxicity profiles of GFD NCs, 4T1 mammary tumor cells and U87 glioblastoma cells were incubated with GFD NCs at varied concentrations (12.5, 6.25, 3.13, 1.56, 0.78, and 0.39 µg ml$^{-1}$) in both acidic (pH = 6.0) and neutral (pH = 7.4) culture mediums for 24 h. The cytotoxicity was evaluated via a cell-counting kit-8 (CCK-8) assay. It has been found that cell viabilities are highly dependent on the dosage of GFD NCs and pH value (Fig. 4a, d). In the 4T1 cytotoxicity assay, the GFD NCs present 44.32, 54.71, 61.29, 64.70, 67.67, and 81.11% cell viabilities at descending concentrations under the acidic condition, while much higher cell viabilities, 67.14, 71.08, 85.48, 99.53, 98.40 and 97.39%, could be observed at corresponding concentrations under neutral condition (Fig. 4a). Similarly, in the U87 cytotoxicity assay under identical concentrations of GFD NCs, the acidic condition induces significantly higher cytotoxicity as compared to that under the neutral condition (Fig. 4b).

Especially, the cytotoxicity toward 4T1 and U87 cancer cells induced by 12.5 µg ml$^{-1}$ of GFD NCs could be rescued by the addition of varied concentrations of typical anti-oxidant agent - L-ascorbic acid (50, 25, 12.5, and 6.25 µg ml$^{-1}$), indicating that the cytotoxicity of GFD NCs towards cancer cells does originate from the oxidative damages by active radicals (Fig. 4c, d). The cytotoxicity profiles of FD NCs and DMSN support under neutral and acidic conditions are also acquired for both cancer cell lines (Supplementary Fig. 5). No significant decreases of cell viabilities in both cases could be observed, indicating that the cytotoxicity of GFD NCs could only be induced by the synergistic and sequential effects of GOD and Fe$_3$O$_4$ NPs rather than by GOD or Fe$_3$O$_4$ NPs alone.

To visually observe the viable and dead cell distributions, 4T1 cancer cells were stained with calcein-AM and PI solution after co-incubation with varied concentrations of GFD NCs (2.5, 5, and 10 µg ml$^{-1}$) under neutral or acidic conditions for 4 h. The viable and dead cells were stained with green and red fluorescence, respectively, which could be visually observed on a confocal laser scanning microscope (CLSM). The CLSM images show that no significant or only minor portion of 4T1 cells are damaged at 2.5 and 5 µg ml$^{-1}$ GFD NCs assayed under neutral and acidic conditions, respectively. A majority of dead cells are observed when incubated with GFD NCs at an elevated concentration (10 µg ml$^{-1}$) under neutral condition. Comparatively, under acidic condition, almost all 4T1 cancer cells are dead at 10 µg ml$^{-1}$ of GFD NCs (Fig. 4e). These observations match well with the cytotoxicity profiles of GFD NCs (Fig. 4a), which is supposed to be caused by the highly toxic hydroxyl radicals in situ generated by sequential GFD NCs. A ROS fluorescence probe 2′,7′-dichlorofluorescin diacetate (DCFH-DA) was further applied to stain 4T1 cells after incubation with 10 µg ml$^{-1}$ of DMSN, FD NCs, and GFD NCs, to demonstrate the production of ROS (Fig. 4f). The green fluorescence is hardly observable in DMSN- and FD NCs-stained cancer cells, indicating the insignificant ROS generation by DMSN and FD NCs. Contrarily, strong green fluorescence can be observed in the CLSM image of GFD NCs-treated cancer cells under pH = 6.0, implying the massive intracellular ROS production. Comparatively, GFD NCs under pH = 7.4 displays much weaker fluorescence intensity than that under pH = 6.0, revealing much less efficient ROS production under neutral condition than acidic one (Fig. 4f). Corresponding fluorescence-intensity profiles were also collected and analyzed (Supplementary Fig. 6), providing the quantitative evaluations on the ROS-generating capability of the as-prepared sequential GFD NCs.

**Biodegradation behavior and biocompatibility of GFD NCs.** The biodegradation and biocompatibility of GFD NCs determine the future possible clinical translation. Therefore, the biodegradation behavior of GFD NCs was initially evaluated under neutral (pH = 7.4) and acidic (pH = 6.0) simulated body fluid (SBF) mediums, which were applied to imitate the in vivo neutral healthy body fluids and intratumoral mildly acidic environment, respectively (Fig. 5a). The structure/morphology evolutions and Si-based biodegradation product concentrations were monitored by TEM observations and ICP-OES measurements, respectively (Fig. 5b, c). Accumulatively, GFD NCs degrade at a higher rate in neutral medium than in acidic one within a degradation period of 15 days, implicating that GFD NCs could provide more durable therapeutic performance in tumorous acidic area (Fig. 5b). The morphology of GFD NCs remained spherical in both neutral and acidic mediums in 1-day degradation. Moderate structural distortion of the silica framework could be observed on GFD NCs in neutral SBF medium, while the framework of GFD NCs shows

much higher stability in acidic SBF medium in 3-day biodegradation. As the degradation time period was prolonged to 5 days, GFD NCs exhibited significant structural collapse and substantial biodegradation in both neutral and acidic mediums. Clearly, dendritic structure could still be observed on the GFD NCs in acidic medium because of the slower biodegradation, while in neutral medium, the framework of the DMSN has totally collapsed, leading to the disappearance of the mesoporous structure and the shrinkage of the particles. The biodegradation became more apparent after another 2 days of degradation under both conditions (Fig. 5c).

Furthermore, the biodegradation behavior and structure evolution of GFD NCs were investigated and in situ observed by bio-TEM directly in 4T1 tumor cells. The bio-TEM images of ultrathin section of 4T1 cancer cells after co-incubation with GFD NCs for varied time intervals (1, 3, 5, and 7 days) were acquired (Supplementary Fig. 7). The GFD NCs could be readily endocytosed into the cytoplasm of the tumor cells and the intact dendritic spheroid morphology could still be observed in 1-day intracellular biodegradation. The dendritic structure has been destructed gradually in 3- and 5-day biodegradations, and no intact spheres but totally destructed dendritic structures could be found in the cancer cells in 7 days of incubation. These results indicate that the as-obtained GFD NCs are intracellularly biodegradable, and expected to be in vivo excreted out of the body after therapeutic functioning.

The in vivo biosafety was evaluated on healthy Kunming mice administrated with saline (control) and GFD NCs at doses of 5, 10, and 20 mg kg$^{-1}$ intravenously. The body weights of the mice were recorded within the whole evaluation period of 30 days, which show that the administration of GFD NCs has no significant in vivo influence toward the mice growth (Supplementary Fig. 8a). The blood urea nitrogen and creatinine (CREA) levels of all experiment groups show no significant variations compared to the control group, indicating that the kidney functions of all mice are normal and healthy (Supplementary Fig. 8b, c). The liver status is reflected by the level of alanine aminotransferase (ALT). The ALT levels of mice in 5 and 10 mg kg$^{-1}$ groups are as low as the control group, while mice in 20 mg kg$^{-1}$ group show a slight upregulation on ALT level, probably due to the higher metabolic burden in this group. Aspartate aminotransferase and alkaline phosphatase levels remain normal for all groups, elucidating that the liver functions of these groups are healthy (Supplementary Fig. 8d). In addition, no obvious difference on blood indexes including hemoglobin, mean corpuscular hemoglobin, mean corpuscular volume, red blood cells, and hematocrit could be observed, proving the uniform biochemical status and no significant infections during the whole evaluation period (Supplementary Fig. 8e–i). The histopathological images of heart, liver, spleen, lung, and kidney for all groups show no observable pathological abnormalities (Supplementary Fig. 9), indicating the high histocompatibility of GFD NCs. These preliminarily but comprehensive in vivo evaluations demonstrate the high biocompatibility of GFD NCs for possible in vivo therapeutic applications.

**Pharmacokinetics of intravenously injected GFD NCs.** The pharmacokinetics of the intravenously injected sequential GFD NCs were assessed to reveal the in vivo behaviors of GFD NCs. Initially, the biodistributions of the nanocatalysts in major organs in the time courses were studied (Fig. 6a). It has been found that GFD NCs are heavily distributed in liver and spleen because of the capture by the reticuloendothelial system. The relative distributed amounts of GFD NCs within tumor are 4.96% in 2 h post injection, 7.12% in 24 h post injection, and finally 6.95% in 48 h

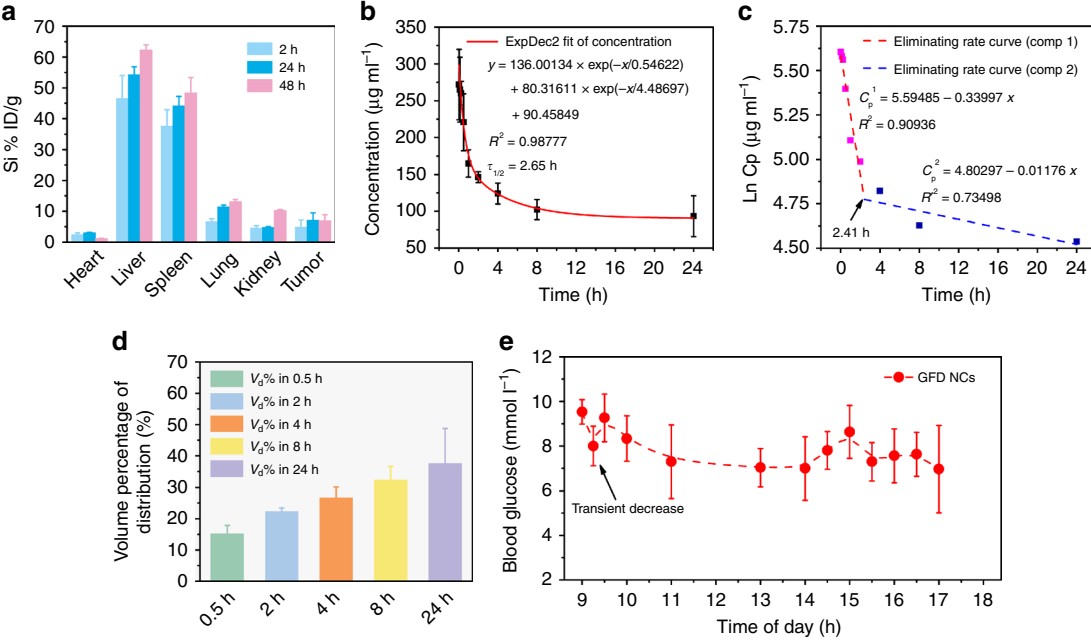

**Fig. 6** Pharmacokinetics and pathoglycemia studies of GFD NCs. **a** The biodistribution of Si (% injected dose (ID) of Si per gram of tissues) in main tissues and tumor in 2, 24, and 48 h of intravenous administrations of GFD NCs ($n = 3$). **b** The blood circulation curve of intravenously injected GFD NCs ($n = 5$). The half-time ($\tau_{1/2}$) was calculated to be ~2.65 h. **c** The eliminating rate curve of intravenously injected GFD NCs from the blood circulation curve according to the ln(concentration)–$T$ relationship. The two stage eliminating rates of GFD NCs can be seen. **d** The apparent volume percentage of distribution ($V_d$%) of intravenously injected GFD NCs ($n = 5$). **e** Blood glucose level of mice in time of day. Mice ($n = 3$) was intravenously injected of GFD NCs at 09:00. Mean values and error bars are defined as mean and s.d., respectively

post injection. Their effective accumulation in tumor is a result of the EPR effect. In the blood-circulation experiment, a circulating half-life of 2.65 h in blood stream was obtained (Fig. 6b). The eliminating rate constants of GFD NCs were calculated to be $-0.340\ \mu g\ ml^{-1}$ per h in the first stage and it decreased to $-0.012\ \mu g\ ml^{-1}$ per h in the second stage with a shifting time interval of 2.41 h (Fig. 6c). The apparent volume percentage of distribution of GFD NCs shows increasing distribution kinetics in the whole blood of body (Fig. 6d). In addition, the blood glucose level of mice after receiving the intravenously injected GFD NCs was also monitored in 8 h. It can be found that in 30 min post injection, the blood glucose exhibits a transient and moderate decrease by GOD moiety from the injected GFD NCs, and the blood glucose level spontaneously recovered in 1 h post injection and no pathoglycemia could be observed within the extended observation time window (Fig. 6e).

**In vivo tumor catalytic therapeutics of sequential GFD NCs.** The intriguing in vitro catalytic-therapeutic efficacy and high biodegradability/biocompatibility of sequential GFD NCs may imply their potentially high in vivo therapeutic outcome. To verify this, the therapeutic performance of GFD NCs was examined on 4T1 mammary tumor xenograft on specific pathogen-free BALB/c nude mice. Saline (control group) and GFD NCs at different doses (5 and 10 mg kg$^{-1}$, therapeutic groups) were administrated intravenously and intratumorally to investigate the therapeutic performances, respectively (Fig. 7a). During 15 days of the therapeutic period, the body weights of mice in control and all therapeutic groups show no significant variations (Fig. 7b), indicating that no significant toxicity has been induced by the injection of GFD NCs during the therapeutic treatment. On tumor suppression assessments, both intravenous and intratumoral administrations present satisfactory suppression effects (Fig. 7c, d). To be more specific, according to the variation of the relative tumor volume, the suppression rates have

been calculated to be 39.75 and 64.67%, respectively at 5 and 10 mg kg$^{-1}$ doses administrated intravenously, while 38.23 and 68.89% at the corresponding GFD NCs doses administrated intratumorally (Fig. 7e). The quantitative antitumor efficacies calculated by the variations of tumor weights show comparable results (Fig. 7e), i.e., 75.95 and 77.60%, respectively at 5 and 10 mg kg$^{-1}$ doses administrated intravenously, and 28.74 and 89.64% at corresponding GFD NCs doses administrated intratumorally. After the therapeutic processes, tumors of mice in all groups are dissected and compared in Fig. 7f, g, which visually demonstrate that 4T1 mammary tumor growth could be effectively suppressed after the administration of GFD NCs both intravenously and intratumorally.

The satisfactory tumor suppression effects are supposed to be contributed by the highly toxic hydroxyl radicals produced by the sequential biological/chemical-catalytic reactions by GFD NCs (Fig. 1b). During this process, the sequential GFD NCs can effectively deplete the glucose nutrients and meanwhile, produce large amounts of $H_2O_2$ for subsequent chemo-catalytic Fenton reaction by $Fe_3O_4$ NPs under the mildly acidic microenvironment of tumor. The as-produced toxic hydroxyl radicals could then kill cancer cells in a mitochondria-mediated apoptosis pathway[39].

To further confirm and generalize our result, the in vivo therapeutic experiments were also conducted on another tumor xenograft (U87 glioblastoma model) following the similar protocol in parallel to the intravenous evaluation on 4T1 mammary cancer model (Fig. 7h). It is clear that the U87 glioblastoma tumor growth could be significantly suppressed similarly (Fig. 7i, j). The tumor-suppression rates were calculated according to the volume (5 mg kg$^{-1}$: 22.61%, 10 mg kg$^{-1}$: 57.24%) and weight (5 mg kg$^{-1}$: 40.05%, 10 mg kg$^{-1}$: 54.79%) changes of dissected tumors (Fig. 7j), further demonstrating the substantial in vivo efficacy of GFD NCs in tumor therapy.

To evaluate the pathological damages to tumors and major organs by GFD NCs, the histopathology images of the dissected

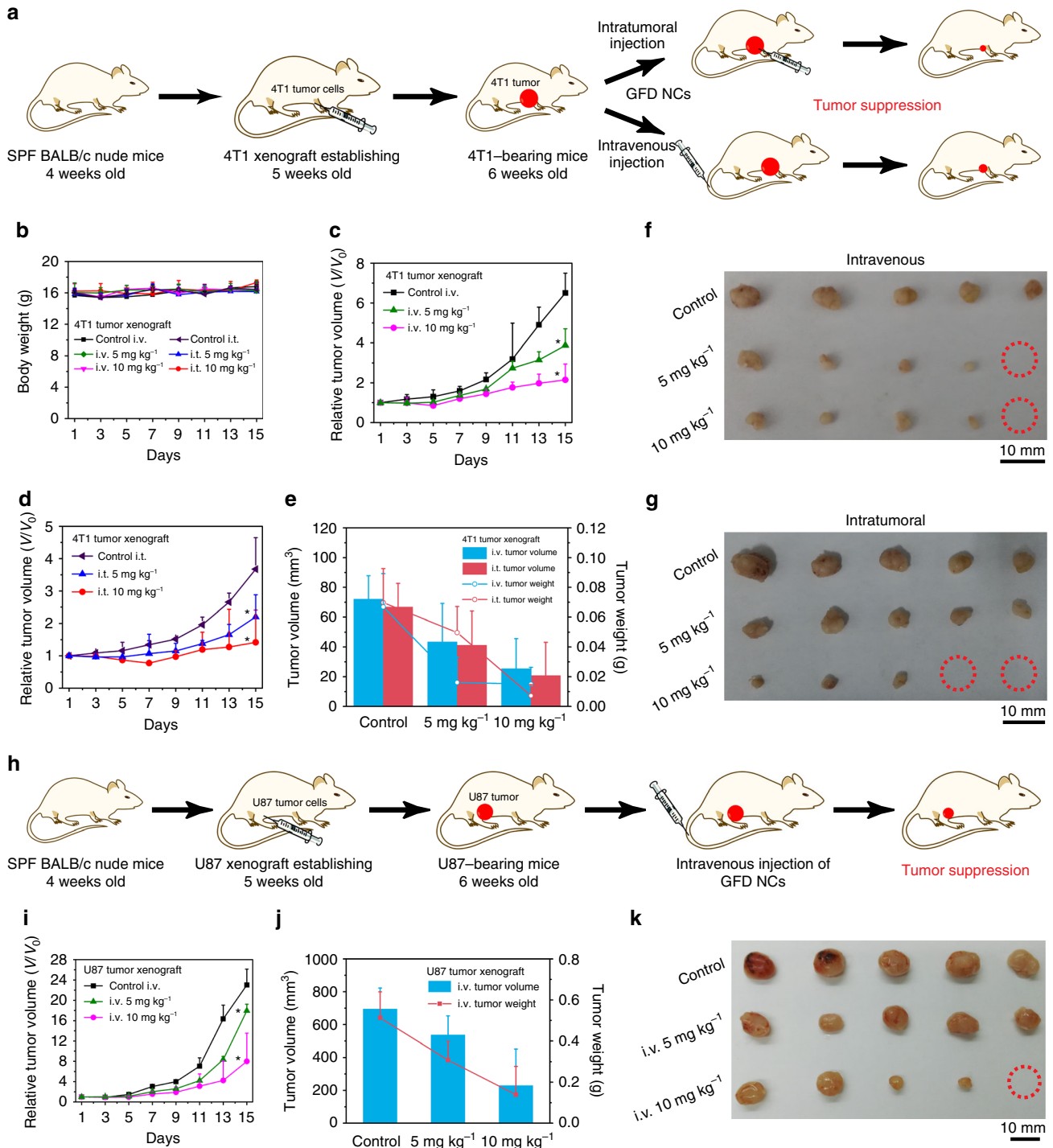

**Fig. 7** In vivo catalytic-therapeutic performance of GFD NCs against 4T1 and U87 tumor xenografts. **a** Schematic illustration of 4T1 tumor xenograft establishment, saline and GFD NCs administration modalities, and therapeutic outcome. **b** The body weights of nude mice bearing 4T1 tumor xenografts after injections with saline (control group) and GFD NCs (5 and 10 mg kg$^{-1}$, therapeutic group) intravenously and intratumorally, respectively. The relative tumor volumes of 4T1 bearing nude mice treated with saline (control group) and different doses of GFD NCs (therapeutic group) via (**c**) intravenous and (**d**) intratumoral modes ($n = 5$). All the above data were collected and measured every 2 days. Statistical significance is assessed by Student's one-sided $t$ test compared to the control group. *$p < 0.05$, **$p < 0.01$. **e** The tumor volumes and tumor weights of dissected tumors from each group after 15 days of therapies. Digital photographs of the dissected tumors from (**f**) intravenous and (**g**) intratumoral groups after 15 days of therapies. *Red circles* indicate the eliminated tumors. **h** Schematic illustration of U87 tumor xenograft establishment, saline and GFD NCs administration modalities, and therapeutic outcome. **i** The relative tumor volumes of U87 bearing nude mice intravenously injected with saline (control group) and GFD NCs (5 and 10 mg kg$^{-1}$, therapeutic group) ($n = 5$). Statistical significance is assessed by Student's one-sided $t$ test compared to the control group. *$p < 0.05$, **$p < 0.01$. **j** The tumor volumes and tumor weights of dissected tumors from each group in 15 days of therapies. **k** Digital photographs of the dissected tumors from each group in 15 days of therapies. *Red circles* indicate the eliminated tumors. Mean values and *error bars* are defined as mean and s.d., respectively. *i.t.* intratumorally; *i.v.* intravenously

4T1 and U87 tumors and major organs are shown in Supplementary Figs. 10–13. The significant destruction of tumor cells could be directly observed in the hematoxylin and eosin staining (H&E) images of 4T1 (Supplementary Fig. 10a) and U87 (Supplementary Fig. 12a) tumor tissues and cells. In the images of terminal deoxynucleotidyl transferase-mediated dUTP-biotin nick end labeling (TUNEL) assays of 4T1 tumors, enhanced cell apoptosis could be observed. Compared to the control group, lower antigen Ki-67 expression in therapeutic groups could be observed. To be more quantitative, Ki-67 indexes were calculated as positive nucleus percentage of total nucleus and plotted in Supplementary Fig. 10d, which presents the much lower Ki-67 index and inhibited proliferation of tumors cell in the therapeutic group. Inhibited cell proliferation and enhanced cell apoptosis could be observed as well in the U87 tumor xenograft immunohistopathologic assay (Supplementary Fig. 12a–c). H&E staining assays of the major organs from all experimental groups show no significant damages when compared to the control groups, further demonstrating the high biocompatibility of the sequential GFD NCs during the therapeutic process (Supplementary Figs. 11 and 13). Additionally, the glucose and oxygen deprivation effects by GOD cannot be completely excluded, which might lead to poor oxygen supply and create the hypoxic regions, consequently affect the therapeutic efficacy of GFD NCs for tumors.

## Discussion

A concept of sequential catalysis has been introduced into nanomedicine for efficient tumor growth suppression by elaborately designing multifunctional nanocatalysts with excellent biodegradability and biocompatibility. As a paradigm, natural GOD and ultrasmall $Fe_3O_4$ NPs were simultaneously loaded into the large pore-sized DMSN to construct sequential GFD NCs. A proof of concept on the sequential catalytic therapeutics of GFD NCs has been successfully demonstrated. The GOD in GFD NCs could effectively deplete the glucose in tumor cells (nutrient deprivation), meanwhile produce considerable amounts of $H_2O_2$ for the subsequent ultrasmall $Fe_3O_4$ NPs-based chemo-Fenton catalysis under the specific acidic microenvironment of tumor, resulting in efficient generation of highly toxic hydroxyl radicals and consequent tumor-cell apoptosis and death. In this sequential catalytic reaction, the Michaelis–Menten steady-state kinetics has been determined as $V_{max} = 4.22 \times 10^{-8}$ M s$^{-1}$ and $K_M = 10.93$ mM, showing the high catalytic performance of GFD NCs. The in vivo therapeutic performance of the biodegradable and biocompatible GFD NCs exhibits highly desired tumor-suppression effect toward both the 4T1 mammary tumor xenograft both intravenously (64.67%) and intratumorally (68.89%) at the dosage of 10 mg kg$^{-1}$, and U87 tumor xenograft (57.24%) intravenously at the same dosage. This work shows that the introduction of elaborately designed nanocatalysts into the tumor tissue can trigger the specific sequential reactions within and in responses to the specific TME to suppress the tumor growth, which provide a promising strategy for the efficient tumor therapy concurrently with largely enhanced tumor specificity and mostly mitigated side effects to normal tissues/organs.

## Methods

**Fabrication of sequential GFD NCs**. The DMSN supports and 2 nm ultra-small $Fe_3O_4$ nanoparticles were synthesized according to the published literatures[26, 27] and redispersed into trichloromethane (CHCl$_3$), respectively. Initially, 2 ml of $Fe_3O_4$ NPs CHCl$_3$ dispersion (5 mg ml$^{-1}$) was added into 20 ml of DMSN CHCl$_3$ dispersion (1 mg ml$^{-1}$) in a dropwise and stirring manner. After 12 h of adsorption, the $Fe_3O_4$@DMSNs nanocatalysts were collected by centrifugation. Afterwards, $Fe_3O_4$@DMSN nanocatalysts (30 mg) were dispersed into 200 ml of methoxy PEG silane-5000 ($M_w = 5000$) ethanol solution (0.5 mg ml$^{-1}$) and magnetic stirring under 60 °C was applied. After 24 h PEGylation process, the PEGylated

$Fe_3O_4$@DMSNs were centrifuged and rinsed with ethanol and deionized water several times to remove the residual PEG. The obtained PEGylated $Fe_3O_4$@DMSNs were dissolved into 10 ml of sodium acetate buffer (NaAc, 3 M, pH = 5.2) under mild magnetic stirring, followed by the addition of GOD (10 mg). After another 12 h collection, the sequential GFD NCs could be acquired by centrifugation.

**Characterization**. Transmission electron microscope (TEM), electron energy loss spectrum (EELS), energy dispersive X-ray spectroscopy (EDS), and corresponding EDS-mapping were adopted for morphology and elemental distribution analysis on JEM-2100F electron microscope operated at 200 kV. DLS was conducted on Malvern Zetasizer Nanoseries (Nano ZS90) for sample zeta potential and hydro-dynamic particle size determination. The nitrogen (N$_2$) adsorption–desorption isothermal curve and corresponding pore-size distribution of DMSN supports were measured by Micromeritics Tristar 3000 system. ESR spectrum of GFD NCs was measured by Bruker EMX1598 spectrometer. 5,5-dimethyl-1-pyrroline-N-oxide (DMPO) was elected as the nitrogen trapping agent to evaluate the hydroxyl radical generation in the nanomedical catalysts. The optical absorbance spectra were conducted on Molecular Device SpectraMax M2.

**Michaelis–Menten kinetics**. One milliliter of 3.2 mM TMB was applied to monitor the chromogenic reaction ($\lambda = 650$ nm) of 100 µg ml$^{-1}$ GFD NCs upon addition of a series $H_2O_2$ or β-D-glucose concentration. NaAc buffer solution (20 mM pH = 5.2) was used to fill and fix the final volume to 3 ml. The Michaelis–Menten kinetic curve of $Fe_3O_4$ NPs and sequential GOD-$Fe_3O_4$ NPs in GFD NCs could be acquired by plotting the respective initial velocities against $H_2O_2$ and D-glucose concentrations. The Michaelis–Menten constant ($K_M$) and maximal velocity $V_{max}$ were calculated via the Lineweaver–Burk plotting.

**Cellular experiments**. The 4T1 mammary and U87 glioblastoma cell lines were obtained from Cell Bank, the Committee of Type Culture Collection of Chinese Academy of Sciences, and are not listed by International Cell Line Authentication Committee as cross-contaminated or misidentified cell lines (v8.0, 2016). These cell lines have passed the conventional tests of cell line quality control methods (e.g., morphology, isoenzymes, and mycoplasma). The 4T1 mammary and U87 glioblastoma cells were cultured in the media of Dulbecco's modified Eagle medium (DMEM), containing 4 mM L-glutamine, 4500 mg l$^{-1}$ glucose, 10% fetal bovine serum (FBS), 100 units per ml streptomycin, and 100 units per ml penicillin, supplied by GE Life Sciences Co. Ltd. The cell lines were incubated at constant temperature of 37 °C and 5% CO$_2$ in the incubator. Cell lines were subcultured for subsequent experiment by adding 0.25% trypsin, 10% FBS, and fresh DMEM high-glucose medium.

For in vitro cytotoxicity assays, 4T1 cancer cells (U87 cancer cells) were inoculated into a 96-well plate with a cell density of 3000 cells per well and incubated for 12 h to allow the attachment of cells. The medium of the 96-well plate was discarded followed by rinsing with phosphate-buffered saline (PBS) twice. Subsequently, DMSNs, FD NCs, and GFD NCs with a serial concentration of 12.5, 6.25, 3.13, 1.56, 0.78, and 0.39 µg ml$^{-1}$ were dispersed into the 10% FBS containing DMEM high-glucose medium and then inoculated into the 96-well plate instead of the PBS solution. The pH value was adjusted to 6.0 by the addition of HCl. After another 24 h incubation at 37 °C, the aforementioned cell mediums were discarded and rinsed with PBS carefully, followed by the addition of 10% Cell Counting Kit-8 (CCK-8 assay) containing DMEM high-glucose medium at 100 µl per well. CCK-8 could selectively stain the viable cell with enough speed, accuracy, and repeatability in 4 h read by a spectrometer at 450 nm.

For in vitro cytotoxicity assays with L-ascorbic acid addition, 4T1 cells (or U87 cells) were inoculated into a 96-well plate with a cell density of 3000 cells per well and incubated for 12 h to allow the attachment of cells. The medium of the 96-well plate was discarded followed by rinsing with PBS twice. Subsequently, GFD NCs were dispersed into the 10% FBS containing DMEM high-glucose medium at the concentration of 12.5 µg ml$^{-1}$ and then inoculated into the 96-well plate instead of the PBS solution. The pH value was adjusted to 6.0 by the addition of HCl. After 4 h incubation at 37 °C, the cell mediums were discarded and replaced with L-ascorbic acid dispersed into 10% FBS containing DMEM high-glucose medium (50, 25, 12.5, 6.25, and 3.125 µg ml$^{-1}$). After another 20 h incubation at 37 °C, the cell mediums were discarded and rinsed with PBS carefully, followed by the addition of 10% CCK-8 containing DMEM high-glucose medium at 100 µl per well. CCK-8 could selectively stain the viable cell with enough speed, accuracy, and repeatability in 4 h read by a spectrometer at 450 nm.

For ROS observations on CLSM, $1 \times 10^5$ of 4T1 cancer cells were digested and resuspended into 1 ml 10% FBS containing DMEM high-glucose medium and subcultured into φ 15 CLSM-exclusive culture disk for further 12 h incubation. Subsequently, the medium was discarded and the disks were rinsed by PBS twice before 1 ml of DMEM high-glucose (at pH 7.4 or 6.0) containing 2.5 µg ml$^{-1}$ of DMSN, FD NCs, and GFD NCs were replaced. Finally, the above medium was removed completely by PBS rinsing followed by the fluorescence probe addition. Herein, 100 mM non-fluorescent 2′,7′-dichlorofluorescin diacetate (DCFH-DA) was applied to reduce ROS for 15 min, forming fluorescent matter DCF ($\lambda_{ex} = 480$ nm, $\lambda_{em} = 525$ nm) which could be observed on confocal laser scanning microscopy (FV 1000, Olympus).

For viable and dead cells observations on CLSM, 4T1 cells were incubated into $\varphi$ 15 CLSM-exclusive culture disk. In order to visualize the viable cells and dead cells after 24 h cytotoxicity, the 3′,6′-Di(O-acetyl)-4′,5′-bis[N,N-bis (carboxymethyl)aminomethyl]fluorescein, tetraacetoxymethyl ester (calcein-AM) / Propidium iodide (PI) staining reagents were applied to stain the viable cells as green fluorescence ($\lambda_{ex}$ = 490 nm, $\lambda_{em}$ = 515 nm) and dead cells as red fluorescence ($\lambda_{ex}$ = 535 nm, $\lambda_{em}$ = 617 nm). Specifically, 100 μl of 20 mM of calcein-AM solution and 100 μl of 20 mM of PI solution were added after the removal of the culture medium and rinsing of the disks. After 15 min of incubation, staining solution were removed and rinsed by PBS twice and the samples could be subsequently visualized by CLSM.

**Animal experiments.** Female Kunming mice and female BALB/c nude mice aged 4 weeks were purchased from Beijing Vital River Laboratory Animal Technology Co., Ltd. All the animal procedures were performed under the protocols approved by Department of Laboratory Animal Science, Fudan University. All animal experiments were in agreement with the guidelines of the Institutional Animal Care and Use Committee of Department of Laboratory Animal Science, Fudan University.

For in vivo biosafety evaluation, female Kunming mice were separated into four groups ($n = 4$) randomly when they were 5 weeks old: control, 5, 10, and 20 mg kg$^{-1}$ groups. Corresponding doses of GFD NCs (saline for control groups) were injected intravenously into the mice and their body weights were measured every 3 days to evaluate the in vivo biosafety. After 30 days of evaluation period, all Kunming mice were sacrificed for anatomy and histopathological analysis.

For in vivo pharmacokinetic evaluation, BALB/c nude mice were randomly assigned into three groups ($n = 3$). They were subcutaneously transplanted with 4T1 mammary cancer cell ($1 \times 10^6$ cells per site) when they were 5 weeks old. The tumors were allowed to grow to 30 mm$^3$ at the start of the experiment. All mice were then intravenously injected with GFD NCs (10 mg kg$^{-1}$). At 2, 24, and 48 h post injection, mice were sacrificed. Major organs (heart, liver, spleen, lung, and kidneys) and tumors were dissected, rinsed with PBS, weighted, and homogenized. The biodistributions in different organs and tumors were calculated as Si percentage of injected dose per gram of tissues. The apparent volumes percentage of distribution ($V_d$%) were calculated as $X/(V \times C_p)$, while $X$ is referred to the total amount of injected drugs (GFD NCs) and $V$ is referred to the volumes of whole blood.

Female Kunming mice ($n = 5$) were intravenously injected with GFD NCs when they were 4 weeks old. At 2, 8, 15, 30 min, 1, 2, 4, 8, and 24 h, 10 μl blood were drawn by nicking the tail vein and dispersed into 990 μl physiological saline contained heparin sodium injection (50 units per ml). The concentrations of Si ($C_p$) were measured by ICP-OES. The in vivo circulating half-life of GFD NCs in blood stream ($\tau_{1/2}$) is calculated by a double-compartment pharmacokinetic model. The eliminating rate curve was conducted by plotting ln($C_p$) against time and fitted according to the two-compartment models. The eliminating rates were reflected by the slopes of the curve.

For blood glucose assays, female Kunming mice ($n = 3$) were intravenously injected with GFD NCs when they were 4 weeks old. At 15, 30 min, 1, 2, 4, 5, 5.5, 6, 6.5, 7, 7.5, 8, and 12 h, blood was collected by nicking and milking the tail vein gently, and placed on the test strip of a commercial glucometer (Accu-chek performa, Roche, Switzerland). This glucometer applies glucose dehydrogenase method to measure the blood glucose.

For in vivo therapeutic evaluation, all BALB/c nude mice were randomly assigned ($n = 5$) into groups. They were subcutaneously transplanted with 4T1 mammary cancer cell ($1 \times 10^6$ cells per site) or U87 cancer cells ($1 \times 10^6$ cells per site) when they were 5 weeks old. The tumors were allowed to grow to 20 mm$^3$ at the start of the treatment. The relative tumor volume is defined as $V_R = V V_0^{-1} \times 100\%$ ($V_0$: tumor volume on the first day). Corresponding doses of GFD NCs (saline for control groups) were injected intravenously and intratumoral into the nude mice, and their body weights and tumor volume were measured every other day to evaluate the therapeutic performance. The injection of GFD NCs was conducted only once. After 15 days of treatment period, all nude mice were sacrificed for anatomy and histopathological analysis.

**Statistic methods.** The significance of the data is analyzed according to a Student's $t$ test: *$p < 0.05$, **$p < 0.01$, and ***$p < 0.001$. The samples/animals were allocated to experimental groups and processed randomly.

**Data availability.** All data are available from the authors upon reasonable request.

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

## Acknowledgements

We greatly acknowledge the financial support from National Key Research and Development Program of China (grant no. 2016YFA0203700), National Nature Science Foundation of China (grant nos. 51672303 and 51132009), Young Elite Scientist Sponsorship Program by CAST (grant no. 2015QNRC001), Natural Science Foundation of Shanghai (grant no. 13ZR1463500), and Youth Innovation Promotion Association (grant no. 2013169).

## Author contributions

Y.C. designated the idea of the present work. Y.C. and J.S. supervised the project and commented on the project. M.H. and L.W. synthesized and characterized the nanocatalysts, performed in vitro and in vivo experiments, and analyzed the data. M.H. wrote the manuscript. All the authors contributed to the discussion during the whole project.

## Additional information

**Competing interests:** The authors declare no competing financial interests.

