## [Peer Review File · Nature Communications]

Reviewers' comments:

Reviewer #1 (Remarks to the Author):

The authors constructed a cascade system by combining glucose oxidase and Fe₃O₄ nanoparticles into dendritic silica nanoparticle (DMSN). Several features are impressive for tumor therapy with this system. First, the sequential reaction from glucose oxidase to Fe₃O₄ nanozyme makes it feasible to consume glucose to generate free radicals for tumor cell destruction. Second, the whole system is pH-responsive and suitable for tumor acidic microenvironment. Third, the dendritic silica nanoparticle is biodegradable with high biocompatibility. These powerful properties make it as an ideal strategy for tumor therapy without any chemotherapeutic drugs. The authors have proved the sequential reactions with in vitro biochemistry analysis and tested its effect and biocompatibility at cellular and in vivo models. More impressively, the efficient tumor therapy was achieved with intravenous and intratumoral administration of the cascade nanosystem. I think the concept is significant to provide a new way for tumor therapy. In addition, all data and figures are well organized and the manuscript is fitful to the style of Nature Communications. Therefore, I recommend it to be accepted for publication on Nature Communications.

1. For glucose oxidase and ultrasmall Fe₃O₄ nanoparticles loading into DMSN, are there any ways to determine the amount of these substances loaded into DMSN?
2. There is no much detailed information for Bio-TEM characterization for GFD NCs in 4T1 cells. For 7 days treatment, does the GFD NCs keep in a same cell?
3. In Figure 6 for in vivo tumor therapy, the data showed in Figure 6 c is not very consistent with the image in Figure 6 h. In Figure 6 h, the difference between 5 mg/kg and 10 mg/kg is not very significant.

Reviewer #2 (Remarks to the Author):

The manuscript of Huo et al, describes the synthesis of nanocatalysts containing the enzyme Glucose Oxidase and iron oxide particles, as a therapeutic approach against glycolytic and acidotic tumors. The proposed model states that the concerted action of glucose oxidase (production of H₂O₂) and Fe₃O₄ would deplete an essential substrate for tumor growth and produce toxic hydroxyl radicals.

There are key biological questions arising from this model that have not been adequately addressed:

- 1) What is the major action site of the particles, the extracellular space or intracellularly? The authors provide TEM pictures showing intracellular uptake of the particles, however, intracellular pH is believed to be maintained to near neutrality even in acidic environments. Hydroxyl radical production and toxicity is shown to be higher at low pH. Is glucose depletion from the growth media sufficient to cause energetic stress and cell death, and/or extracellular radical species to damage the cell membrane?
- 2) On a similar note, proper controls of in vitro viability (fig4) should include two additional conditions: particles with Glucose Oxidase only, and also supplementation of antioxidants to test if that rescues viability and growth. It is also important that GOD-only particles be tested for tumor growth inhibition (fig6).
- 3) In addition to consuming glucose, glucose oxidase consumes oxygen. Glucose deprivation and acidosis are often found in hypoxic areas to tumors, so it is possible that, in addition to the proposed mechanisms, treatment with GFD NC depletes the already poor oxygen supply and drives hypoxic regions into anoxia and death. That possibility should be given consideration in the discussion and in the proposed model.

- 4) The PK/PD of the particles in mice require some expansion, particularly since it is not well established where the tumor specificity is coming from. Has particle uptake been observed in cells of healthy tissues? Are they trapped by the reticuloendothelial system (RES)? How long do they circulate in the blood?
- 5) More experimental details on the tumor growth experiments are needed. How many 4T1 cells were injected? Subcutaneously or orthotopically? How large were the tumors at the start of treatment? What was the treatment schedule, daily for 15 days? What statistics methods were used?
- 6) Have the experiments been performed in cells other than 4T1? Inclusion of a second cell line would strengthen the generalization of the results.
- 7) Fig 6f shows the final 4T1 tumor weights of particle-treated mice and the largest PBS-treated group is around 60mg. That is considerably smaller than the typical starting volume for xenograft treatment (100-300mg) and difficult to measure accurately. Is that number correct or a typographical error? Similarly small volumes are shown in 6e.
- 8) The IHC images for TUNEL and Ki67 are hard to evaluate, particularly the Ki67. Quantitation of the staining areas would be helpful.

Reviewer #3 (Remarks to the Author):

This manuscript provides a catalytic nanomedical therapeutic concept for tumor therapy, based on the principle of nanozyme to catalyse the generation of ROS in situ in response to the specific acidic condition of tumour microenvironment. This work is novel and interesting. However, before publication in Nature communication, the authors need to add more data to prove its effectiveness and practicability.

Major comments:

1. For tumor selectivity as it is stated in the title "Tumor-Selective Catalytic Nanomedicine by Nanocatalyst Delivery", more evidences in vitro and in vivo are needed to show that the "Nano catalysts" have tumor selectivity. For instance, to prove the ability of GFD NCs specifically target to 4T1 tumors in vivo by intravenously injection (via EPR effects?).
2. The authors stated that one of the major feature of their composite nano-catalyst is its high biocompatibility. However, to prove this, only measuring the body weights of experimental animal and the biodegradation behavior of the materials is not enough. The authors need to add more data, such as the bio-distribution of GFD NCs in vivo after intravenous injection.
3. Requires more data regarding whether or not the glucose oxidase in the composite nanocatalyst could affect the glucose metabolism of the experiment animal. For instance, if the intravenously injection of GFD NCs would induce pathoglycemia?
4. Since the mechanism is still unclear, a direct evidence showing the production of hydroxyl radicals in vivo is needed.

Minor comments:

1. After intravenous injection, the GFD NCs could non-specifically concentrate in certain healthy organs, although the pH of environment is neutral, the GFD NCs still produces abundance of H₂O₂, which could disturb the balance of the redox system?
2. The schematic illustration of Figure 6a is incorrect. The 4T1 tumor model is subcutaneously injected into the mice, not via tail-vein injection.

Response to Reviewer 1

Comments and suggestions from Reviewer 1.

The authors constructed a cascade system by combining glucose oxidase and Fe_3O_4 nanoparticles into dendritic silica nanoparticle (DMSN). Several features are impressive for tumor therapy with this system. First, the sequential reaction from glucose oxidase to Fe_3O_4 nanozyme makes it feasible to consume glucose to generate free radicals for tumor cell destruction. Second, the whole system is pH-responsive and suitable for tumor acidic microenvironment. Third, the dendritic silica nanoparticle is biodegradable with high biocompatibility. These powerful properties make it as an ideal strategy for tumor therapy without any chemotherapeutic drugs. The authors have proved the sequential reactions with *in vitro* biochemistry analysis and tested its effect and biocompatibility at cellular and *in vivo* models. More impressively, the efficient tumor therapy was achieved with intravenous and intratumoral administration of the cascade nanosystem. I think the concept is significant to provide a new way for tumor therapy. In addition, all data and figures are well organized and the manuscript is fitful to the style of Nature Communications. Therefore, I recommend it to be accepted for publication on Nature Communications.

Response: Thank you very much for the positive comment and recommendation. Please find the following detailed responses to your comments and suggestions.

1. For glucose oxidase and ultrasmall Fe_3O_4 nanoparticles loading into DMSN, are there any ways to determine the amount of these substances loaded into DMSN?

Response: Thank you for the question. The loading amount of Fe_3O_4 nanoparticles and GOD in GOD- Fe_3O_4 @DMSN Nanocatalysts (GFD NCs) has been determined by ICP-OES and thermogravimetric measurements, respectively. The results show that the loading amounts of Fe_3O_4 and GOD into GFD NCs are 15.87% and 16.61%, respectively (**Figure S4e-f**).

2. There is no much detailed information for Bio-TEM characterization for GFD NCs in 4T1 cells. For 7 days' treatment, does the GFD NCs keep in a same cell?

Response: Thank you for the question. In this work, bio-TEM was used to directly observe the structure evolution of GFD NCs in cancer cells. At the initial stage, a large number of GFD NCs were found in the cancer cells with relatively intact structure (1 d). The structure collapse of GFD NCs was then observed in cells during the prolonged degradation time period (3 to 5 days). Only slight signs, such as small cracks, of GFD NCs were observed in 7 d co-incubation (**Figure S7**). Bio-TEM characterization only provides the general information of the biodegradation process of GFD NCs in cells, and the further quantitative evaluation has been made by ICP tests (**Figure 5b**). It is indeed not practical to directly observe the biodegradation of GFD NCs in the same cell after the co-incubation for largely varied durations, as all cells at each time point of incubation should be harvested and fixed for bio-TEM characterization.

3. In Figure 6 for *in vivo* tumor therapy, the data showed in Figure 6c is not very consistent with the image in Figure 6h. In Figure 6h, the difference between 5 mg/kg and 10 mg/kg is not very significant.

Response: Thank you for the question. The original Figure 6c (**Figure 7c** in the revised manuscript) was measured by caliper when the mice were alive (*in vivo*), while the original Figure 6h (**Figure 7f** in the revised manuscript) was obtained by dissecting tumors from the mice (*ex vivo*). Deviations must be present in these measurements and calculations of the tumor volume *in vivo* and *ex vivo*.

Response to Reviewer 2

Comments and suggestions from Reviewer 2.

The manuscript of Huo et al, describes the synthesis of nanocatalysts containing the enzyme Glucose Oxidase and iron oxide particles, as a therapeutic approach against glycolytic and acidotic tumors. The proposed model states that the concerted action of glucose oxidase (production of H_2O_2) and Fe_3O_4 would deplete an essential substrate for tumor growth and produce toxic hydroxyl radicals.

Response: Thank you very much for the kind comments and suggestions. Please find the following detailed responses.

There are key biological questions arising from this model that have not been adequately addressed:

1. What is the major action site of the particles, the extracellular space or intracellularly? The authors provide TEM pictures showing intracellular uptake of the particles, however, intracellular pH is believed to be maintained to near neutrality even in acidic environments. Hydroxyl radical production and toxicity is shown to be higher at low pH. Is glucose depletion from the growth media sufficient to cause energetic stress and cell death, and/or extracellular radical species to damage the cell membrane?

Response: Thank you very much for the comments. We believe that GFD NCs will act as the effective cascade catalyst where both glucose and acidity are present, therefore, the major action site of the developed GFD NCs is in the intracellular lysosome during the cellular level evaluation. It is well-known that nanoparticles are typically endocytosized into cancer cells, which can further enter the lysosome of cells. The lysosome is typically acidic, which can trigger the cascade reaction to induce the cancer-cell death. However, during the *in vivo* evaluation, both the extracellular and intracellular microenvironments are the major action sites of GFD NCs. It has been widely revealed that the tumor microenvironment is acidic because of the active

intratumoral glycolysis which produces a large amount of lactic acid. Such a unique acidic microenvironment can trigger the sequential reaction to generate hydroxyl radicals by using accumulated GFD NCs as the catalyst. Further intracellular uptake of GFD NCs by cancer cells further induces the cancer-cell death due to the induced catalytic reactions. The related references regarding the acidic microenvironment of cancer cells have been cited in the revised manuscript (Ref. 34 and 35). According to the cytotoxicity assays, the pH value showed critical influence on the catalytic activity of GFD NCs. The profound hydroxyl-radical productions can be clearly observed under acidic condition (pH = 5.5-6.0) rather than neutral condition (pH = 7.4), demonstrating the high therapeutic efficiency of GFD NCs in acidic environment (**Figure 4**).

To evaluate glucose-deprived damage towards the tumor cells, we have assessed the cytotoxicity of GOD@DMSN Nanocatalysts (GD NCs, without Fe₃O₄ loading) against 4T1 and U87 cancer cell lines. It has been found that the viabilities of both 4T1 and U87 cancer cells are unaffected by the glucose-depriving GD NCs, which indicates that the glucose-only depletion from the growth media is insufficient to cause energetic stress and cell death (**Figure S5a-b**).

2. On a similar note, proper controls of in vitro viability (Figure 4) should include two additional conditions: particles with Glucose Oxidase only, and also supplementation of antioxidants to test if that rescues viability and growth. It is also important that GOD-only particles be tested for tumor growth inhibition (Figure 6).

Response: Thank you for the constructive suggestion. According to the reviewer's suggestion, the cytotoxicity profiles of GOD@DMSN Nanocatalysts (GD NCs) against 4T1 and U87 cancer cells have been assessed. It has been found that GD NCs have no significant influence on the viabilities of both 4T1 and U87 cancer cells (**Figure S5**).

In addition, the cytotoxicity assessment by introducing antioxidants was further conducted according to the reviewer's suggestion. Herein the biocompatible ascorbic acid was used as the antioxidant agent. It has been found that the addition of ascorbic acid as an antioxidant could significantly rescue the viabilities of 4T1 and U87 cancer cells (**Figure 4c and d**), further

confirming the oxidative damaging effect of GFD NCs towards cancer cells. Additionally, according to the non-toxic profiles of GOD@DMSN Nanocatalysts (**Figure S5**), it is reasonable to infer that the GOD@DMSN Nanocatalysts would not affect the tumor growth.

3. In addition to consuming glucose, glucose oxidase consumes oxygen. Glucose deprivation and acidosis are often found in hypoxic areas to tumors, so it is possible that, in addition to the proposed mechanisms, treatment with GFD NC depletes the already poor oxygen supply and drives hypoxic regions into anoxia and death. That possibility should be given consideration in the discussion and in the proposed model.

Response: Thank you very much for the constructive suggestion. We fully agree with the reviewer's opinion that the glucose deprivation will possibly induce poor oxygen supply and drive hypoxic regions into anoxia and death. According to the *in vitro* cytotoxicity assays of GOD@DMSN (without Fe₃O₄ NPs) towards cancer cells (4T1 and U87 cancer cell lines), it has been found that solely depleting oxygen from hypoxic tumor cells could not significantly induce the death of cancer cells (**Figure S5a-b**), probably due to the incomplete deoxygenation by the GOD@DMSN. However, such a possibility cannot be totally excluded because of the complicated *in vivo* microenvironment of tumor. Therefore, we have added the related discussion in the revised manuscript according to the reviewer's kind suggestion.

4. The PK/PD of the particles in mice require some expansion, particularly since it is not well established where the tumor specificity is coming from. Has particle uptake been observed in cells of healthy tissues? Are they trapped by the reticuloendothelial system (RES)? How long do they circulate in the blood?

Response: Thank you very much for constructive suggestions and questions. According to the reviewer's suggestion, the PK/PD-related *in vivo* evaluations of GFD NCs have been further conducted, including bio-distribution, blood-circulation and related calculation/discussion. It has been found that GFD NCs can be rapidly captured by reticuloendothelial system (RES) as

revealed by the accumulation in liver and spleen. This is a common phenomenon in nanomedicine. Importantly, these GFD NCs could accumulate into the tumor by the typical enhanced permeability and retention (EPR) effect. The relative accumulation amount of the nanocatalysts in the tumor increased from 4.96% in 2 h post-injection to 7.12% in 24 h post-injection and finally to 6.95% in 48 h post-injection (**Figure 6a**). Therefore, the tumor specificity of GFD NCs is considered to be from the EPR effect and more significantly, the intratumoral mild acidic microenvironment-triggered sequential reaction for the generation of toxic radicals.

In the blood-circulation experiment, a half-life of 2.65 h was measured for GFD NCs circulating in the blood stream. The elimination rate constants of GFD NCs were calculated to be $-0.340 \mu\text{g mL}^{-1} \text{h}^{-1}$ in the first compartment while it decreased to $-0.012 \mu\text{g mL}^{-1} \text{h}^{-1}$ in the second compartment with a shifting time of 2.41 h. The apparent volume percentage of distribution of GFD NCs showed increasing distributing kinetics in the body (**Figure 6b-d**).

5. More experimental details on the tumor growth experiments are needed. How many 4T1 cells were injected? Subcutaneously or orthotopically? How large were the tumors at the start of treatment? What was the treatment schedule, daily for 15 days? What statistics methods were used?

Response: Thank you very much for the suggestions. Details on tumor xenograft establishment and therapeutic treatment were supplemented according to the reviewer's kind suggestion. 4T1 cancer cell (1×10^6 cells/site) and U87 cancer cell (1×10^6 cells/site) were subcutaneously transplanted into 5-weeks-old BALB/c nude mice. The tumor xenografts were allowed to develop to 20 mm^3 before the start of the treatments. The intravenous injection of GFD NCs was conducted only once for each mouse. Body weights and tumor sizes were recorded every other day for 15 days. The relative tumor volume was defined as $V_R = V/V_0 * 100\%$ (V_0 : Tumor volume on the first day). Statistic data are represented as mean \pm s. d. Error bars indicate the standard derivations. The significance of the data was analyzed according to a Student's t test: *p < 0.05, **p < 0.01 and ***p < 0.001.

6. Have the experiments been performed in cells other than 4T1? Inclusion of a second cell line would strengthen the generalization of the results.

Response: Thank you very much for the constructive questions. According to the reviewer's suggestion, we further conducted the experiment on the evaluation of the therapeutic efficiency of GFD NCs on glioma U87 cancer cells, in addition to the breast 4T1 cancer cells. Both *in vitro* and *in vivo* systematic evaluations have been conducted. It has been found that these as-designed GFD NCs are also highly effective in the treatment of glioma U87 cancer cells and their corresponding tumor xenografts in mice (**Figure 4b and d, Figure 7h-k**).

7. Fig 6f shows the final 4T1 tumor weights of particle-treated mice and the largest PBS-treated group is around 60 mg. That is considerably smaller than the typical starting volume for xenograft treatment (100-300 mg) and difficult to measure accurately. Is that number correct or a typographical error? Similarly, small volumes are shown in 6e.

Response: Thank you very much for the kind question. In the *in vivo* therapeutic experiments on 4T1 tumor xenograft, the starting tumor volumes were set at around 20 mm³. During the whole assessment period, we used the calibrated digital vernier caliper to measure the tumor volume, which provided high enough accuracy in measuring the maximum length and width of tumors for such a small tumor volume. The following evaluations have demonstrated the high therapeutic efficiency of GFD NCs based on the suppressed tumor volume as compared to control group. Furthermore, the tumor weights can be much more accurately measured by analytical balance, which demonstrates the highly corresponding results of tumor growth suppression. Therefore, the *in vivo* evaluation using small starting tumor volume in this work is also feasible and effective.

8. The IHC images for TUNEL and Ki67 are hard to evaluate, particularly the Ki67. Quantitation of the staining areas would be helpful.

Response: Thank you very much for the constructive suggestion. According to the reviewer's suggestion, the antigen Ki-67 quantitation of staining areas was performed on ImageJ software using Immunohistochemistry (IHC) Image Analysis Toolbox. The Ki-67 index was calculated from the positive nucleus percentage of total nucleus, and the data have been provided in the revised manuscript (**Figure S10d**).

Response to Reviewer 3

Comments and suggestions from Reviewer 3.

This manuscript provides a catalytic nanomedical therapeutic concept for tumor therapy, based on the principle of nanozyme to catalyse the generation of ROS in situ in response to the specific acidic condition of tumour microenvironment. This work is novel and interesting. However, before publication in Nature communication, the authors need to add more data to prove its effectiveness and practicability.

Response: Thank you very much for the kind comments and suggestions. Please find the following detailed responses.

Major comments:

1. For tumor selectivity as it is stated in the title “Tumor-Selective Catalytic Nanomedicine by Nanocatalyst Delivery”, more evidences in vitro and in vivo are needed to show that the “Nano catalysts” have tumor selectivity. For instance, to prove the ability of GFD NCs specifically target to 4T1 tumors in vivo by intravenous injection (via EPR effects?).

Response: Thank you very much for the constructive suggestion and question. The “tumor specificity” in this work is based on two contributions. First, as widely known, these as-synthesized GFD NCs are capable of accumulating into tumor tissue *via* the typical EPR effect, which has been demonstrated by the *in vivo* biodistribution assessment after intravenous administration of GFD NCs as suggested by the reviewer (**Figure 6a**). The relative accumulation amount of nanocatalysts in tumor increased from 4.96% in 2 h post-injection to 7.12% in 24 h post-injection and finally to 6.95% in 48 h post-injection. In the blood-circulation experiment, a circulating half-life of 2.65 h was calculated for GFD NCs in the blood stream (**Figure 6b**). But more importantly and especially, the as constructed GFD NCs can specifically response to the intratumoral mild acidic microenvironment to initiate cascade catalytic effect, which in situ produces large amounts of toxic oxidative radicals to accomplish the therapeutic function. It is

clear that such a catalytic therapeutic effect is absent in normal neutral tissues.

2. The authors stated that one of the major feature of their composite nano-catalyst is its high biocompatibility. However, to prove this, only measuring the body weights of experimental animal and the biodegradation behavior of the materials is not enough. The authors need to add more data, such as the bio-distribution of GFD NCs *in vivo* after intravenous injection.

Response: Thank you very much for the constructive suggestion. In addition to the measurements of the body-weight changes and biodegradation behaviors of GFD NCs, the *in vivo* biosafety assay was evaluated on healthy Kunming mice administrated with saline (control) and GFD NCs at elevated doses (5 mg/kg, 10 mg/kg and 20 mg/kg) intravenously. The blood indexes and histocompatibility have been systematically evaluated *in vivo*, which further proves the high biocompatibility of GFD NCs (**Figure S8 and S9**). Further, according to the reviewer's kind suggestion, the bio-distribution and blood-circulation assessments have been conducted, and the data have been provided in the revised manuscript (**Figure 6a-c**).

3. Requires more data regarding whether or not the glucose oxidase in the composite nanocatalyst could affect the glucose metabolism of the experiment animal. For instance, if the intravenously injection of GFD NCs would induce pathoglycemia?

Response: Thank you very much for the reviewer's constructive suggestion. According to the reviewer's suggestion, the blood glucose after the intravenous administration of GFD NCs was monitored. It has been found that the blood glucose in mice was transiently decreased in as short as 30 minutes after the intravenous administration, but it then quickly recovered to the normal level. Overall, the blood glucose level has no significant change after the administration of GFD NCs in long term, excluding the possibility of inducing pathoglycemia. The related data have been provided in the revised manuscript (**Figure 6e**).

4. Since the mechanism is still unclear, a direct evidence showing the production of hydroxyl radicals *in vivo* is needed.

Response: Thank you very much for the constructive suggestion. We totally agree with the reviewer's opinion that the direct *in vivo* evidence on the production of hydroxyl radicals is helpful for revealing the mechanism. Actually, we have made several attempts, but unfortunately, we have found that it is extremely difficult to capture these hydroxyl radicals *in vivo* because, as well-known, the produced hydroxyl radicals have very short lifetime while tumor dissection needs much longer time. To date, there are very few direct characterization methods for hydroxyl radicals *in vivo*, as far as we know. Alternatively, we have conducted the extensive *in vitro* characterizations to demonstrate the instant and large productions of hydroxyl radicals, including the intracellular DCFH-DA characterization experiments (**Figure 4f**), ESR spectroscopy (**Figure 3a**) and TMB chromogenic reaction (**Figure 3b-g**), *in vitro*. These results convince us that the high *in vivo* tumor-suppressing effect in this work originates from the production of hydroxyl radicals.

Minor comments:

1. After intravenous injection, the GFD NCs could non-specifically concentrate in certain healthy organs, although the pH of environment is neutral, the GFD NCs still produces abundance of H₂O₂, which could disturb the balance of the redox system?

Response: Thank you very much for the kind question. We agree with the reviewer's opinion that the GFD NCs could accumulate in healthy organs, as demonstrated by the *in vivo* biodistribution results (**Figure 6d**). The neutral environment of these health organs, however, will not trigger the reaction on the production of toxic hydroxyl radicals. Therefore, they are safe to these healthy organs. The production of H₂O₂ in these organs is unavoidable, but our results have shown that these GFD NCs will not induce the toxicity to these healthy organs, as demonstrated by the blood-index tests and *in vivo* histocompatibility assay (**Figure S8-S9**). Therefore, it is considered that the production of H₂O₂ will not disturb the balance of redox system.

2. The schematic illustration of Figure 6a is incorrect. The 4T1 tumor model is subcutaneously injected into the mice, not via tail-vein injection.

Response: Thank you very much for pointing out this issue. It has been corrected accordingly in the revised manuscript (**Figure 7a**).

REVIEWERS' COMMENTS:

Reviewer #1 (Remarks to the Author):

The authors have well improved the manuscript with major revision and made corresponding explanations to the questions from reviewers. The pharmacokinetics data showed the biodistribution of GFD NCs in the body and clarified the influence on the blood glucose level after intravenously injecting GFD NCs. In addition, the antitumor effect was re-approved with another tumor xenograft (U87 glioma cancer), showing the generality of this system for cancer therapy. Based on the improvement and updated information, I think the manuscript is now qualified for the publication. Therefore, I recommend it to be accepted by Nature Communications.

Reviewer #2 (Remarks to the Author):

The additional data have strengthened the claims of the manuscript.
Minor comments: the text would benefit from thorough language editing. A list of instances requiring corrections is following, although the list is not exhaustive.

105-111 "vast expressions" "concentrated glycolytic process" 110: "pyruvate molecules will be exclusively catalyzed". There is measurable remaining mitochondrial oxidation happening in glycolytic or hypoxic tumors.

256: stained with green and red fluorescences - not plural

268: ROS generations - not plural

304: endocytosized

311: elevated- meaning increasing?

320: metabolic potentials ?

348: significant pathoglycemia phenomenon

362: present well dose-dependent suppression effects

369: meanwhile

395 KI-67 expressions

404-406: "Additionally... tumors". Glucose deprivation per se does not lead to poor oxygen supply, however, the glucose oxidase reaction consumes oxygen, increasing oxygen demand. That may exacerbate pre-existing hypoxia.

409: sequential-catalytic concept

414: demonstrated and proved

415: glucose nutrients

529 transfer / transferred

Reviewer #3 (Remarks to the Author):

The authors have already addressed all my concerns. I recommend it to be accepted for publication on Nature Communications.

Response to Reviewer 1

Comments and suggestions from Reviewer 1.

The authors have well improved the manuscript with major revision and made corresponding explanations to the questions from reviewers. The pharmacokinetics data showed the biodistribution of GFD NCs in the body and clarified the influence on the blood glucose level after intravenously injecting GFD NCs. In addition, the antitumor effect was re-approved with another tumor xenograft (U87 glioma cancer), showing the generality of this system for cancer therapy. Based on the improvement and updated information, I think the manuscript is now qualified for the publication. Therefore, I recommend it to be accepted by Nature Communications.

Response: Thank you very much for the recommendation.

Response to Reviewer 2

Comments and suggestions from Reviewer 2.

The additional data have strengthened the claims of the manuscript. Minor comments: the text would benefit from thorough language editing.

Response: Thank you for the kind suggestion. We have further carefully polished the language of the revised manuscript.

Please find the following detailed responses.

110: “pyruvate molecules will be exclusively catalyzed”. There is measurable remaining mitochondrial oxidation happening in glycolytic or hypoxic tumors.

Response: Thank you for the kind reminding. The sentence “pyruvate molecules will be exclusively catalyzed” has been changed to “most pyruvate molecules will be catalyzed” (In Page 5).

404-406: "Additionally... tumors". Glucose deprivation per se does not lead to poor oxygen supply, however, the glucose oxidase reaction consumes oxygen, increasing oxygen demand. That may exacerbate pre-existing hypoxia.

Response: Thank you for the kind reminding. The statement has been changed to “Additionally, the glucose and oxygen deprivation effect by GOD cannot be completely excluded, which might lead to poor oxygen supply and create the hypoxic regions, consequently affect the therapeutic efficacy of GFD NCs for tumors.” (In Page 17).

A list of instances requiring corrections is following, although the list is not exhaustive.

Response: Thank you for pointing out these expression issues. Below please find the table

addressing these language corrections.

NO.	REVIEWER SUGGESTION	CORRECTIONS	PAGE
1	105-111: vast expressions	overexpression	5
2	105-111: concentrated glycolytic process	promoted glycolysis	5
3	256: stained with green and red fluorescences - not plural.	stained with green and red fluorescence	11
4	268: ROS generations - not plural.	ROS generation	12
5	304: endocytosized	endocytosed	13
6	311: elevated- meaning increasing?	at doses of 5 mg kg ⁻¹ , 10 mg kg ⁻¹ and 20 mg kg ⁻¹ intravenously	14
7	320: metabolic potentials?	metabolic burden	14
8	348: significant pathoglycemia phenomenon.	pathoglycemia	15
9	362: present well dose-dependent suppression effects.	present satisfactory suppression effects.	16
10	369: meanwhile.	- deleted	16
11	395: KI-67 expressions.	Ki-67 expression	17
12	409: sequential-catalytic concept.	A concept of sequential catalysis	18
13	414: demonstrated and proved	demonstrated	18
14	415: glucose nutrients.	glucose	18
15	529: transfer / transferred	- deleted	23

Response to Reviewer 3

Comments and suggestions from Reviewer 3.

The authors have already addressed all my concerns. I recommend it to be accepted for publication on Nature Communications.

Response: Thank you very much for the recommendation.